

# Molecular evidence for sex reversal in wild populations of green frogs (*Rana clamitans*)

Max R. Lambert[1], Tien Tran[2], Andrzej Kilian[3], Tariq Ezaz[4] and David K. Skelly[2]

[1] Department of Environmental Science, Policy, and Management, UC Berkeley, Berkeley, CA, USA
[2] School of Forestry and Environmental Studies, Yale University, New Haven, CT, USA
[3] Diversity Arrays Technology, Bruce, ACT, Australia
[4] Institute for Applied Ecology, University of Canberra, Bruce, ACT, Australia

## ABSTRACT

In vertebrates, sex determination occurs along a continuum from strictly genotypic (GSD), where sex is entirely guided by genes, to strictly environmental (ESD), where rearing conditions, like temperature, determine phenotypic sex. Along this continuum are taxa which have combined genetic and environmental contributions to sex determination (GSD + EE), where some individuals experience environmental effects which cause them to sex reverse and develop their phenotypic sex opposite their genotypic sex. Amphibians are often assumed to be strictly GSD with sex reversal typically considered abnormal. Despite calls to understand the relative natural and anthropogenic causes of amphibian sex reversal, sex reversal has not been closely studied across populations of any wild amphibian, particularly in contrasting environmental conditions. Here, we use sex-linked molecular markers to discover sex reversal in wild populations of green frogs (*Rana clamitans*) inhabiting ponds in either undeveloped, forested landscapes or in suburban neighborhoods. Our work here begins to suggest that sex reversal may be common within and across green frog populations, occurring in 12 of 16 populations and with frequencies of 2–16% of individuals sampled within populations. Additionally, our results also suggest that intersex phenotypic males and sex reversal are not correlated with each other and are also not correlated with suburban land use. While sex reversal and intersex are often considered aberrant responses to human activities and associated pollution, we found no such associations here. Our data perhaps begin to suggest that, relative to what is often suggested, sex reversal may be a relatively natural process in amphibians. Future research should focus on assessing interactions between genes and the environment to understand the molecular and exogenous basis of sex determination in green frogs and in other amphibians.

# INTRODUCTION

Sex-determining (SD) modes occur along a continuum bounded by genotypic sex determination (GSD), where phenotypic sex is entirely controlled by genes on sex

Corresponding author
Max R. Lambert,
lambert.mrm@gmail.com

chromosomes, to environmental sex determination (ESD), where environmental conditions (e.g., temperature) determine sex in the absence of any genetic sexual predisposition (*Sarre, Georges & Quinn, 2004*; *Sarre, Ezaz & Georges, 2011*; *Grossen, Neuenschwander & Perrin, 2011*; *Bachtrog et al., 2014*; *Capel, 2017*). Modes along this continuum combine GSD and environmental effects (EE) where some individuals are genetically-predisposed to develop as a given sex and can undergo environmentally-mediated sex reversal and develop their phenotypic sex opposite their genotypic sex (termed GSD + EE in *Valenzuela, Adams & Janzen, 2003*; *Valenzuela et al., 2014*; *Grossen, Neuenschwander & Perrin, 2011*). Vertebrate sex determination displays strong taxonomic patterning. Fish and non-avian reptiles show repeated transitions between GSD, ESD, and GSD + EE whereas GSD is the rule in both mammals and birds (*Sarre, Georges & Quinn, 2004*; *Sarre, Ezaz & Georges, 2011*; *Bachtrog et al., 2014*; *Capel, 2017*).

Perceptions of amphibian sex determination have shifted over time. At the turn of the 20th century, biologists believed amphibian sex determination to be the result of interactions between innate (i.e., genetic) forces and environmental conditions (*King, 1909*, *1919*; *Witschi, 1929*). By the middle of the 20th century, after karyotyping myriad vertebrate sex chromosomes, biologists concluded that the environment was the dominant determiner of amphibian sex and that the genetic basis to sex was weak or non-existent (*Ohno, 1967*). In recent decades, a dearth of evidence from wild populations has led scientists to conclude that amphibians have strict GSD, with environmentally-mediated sex determination considered an aberrant response to extreme temperatures or contamination (*Hayes, 1998*; *Nakamura, 2009*; *Sarre, Ezaz & Georges, 2011*; *Evans, Alexander Pyron & Wiens, 2012*; *Bachtrog et al., 2014*; *Capel, 2017*). We note, however, that emerging theory suggests that sex reversal may be an important process for amphibian evolution (*Perrin, 2009*; *Grossen, Neuenschwander & Perrin, 2012*).

Conclusions about amphibian sex determination have predominantly been drawn from laboratory experiments which have identified putative mechanisms and patterns of environmental sex reversal in amphibians mediated by temperature (*Witschi, 1914*, *1929*, *1930*; *Hsu, Yu & Liang, 1971*, *Dournon et al., 1984*; *Lambert et al., 2018*) or natural and anthropogenic chemicals (*Lambert et al., 2017*; *Hayes et al., 2002*; *Pettersson & Berg, 2007*; *Hermelink et al., 2010*; *Lambert, 2015*; *Lambert, Skelly & Ezaz, 2016*; *Tamschick et al., 2016a*, *2016b*). By contrast, the documentation of discordances between phenotypic and genotypic sexes has not been well studied across populations of any amphibian taxon in the wild, particularly in contrasting ecological contexts.

The only *direct* evidence (i.e., discordance between phenotypic and genotypic sexes) for environmental sex reversal in wild amphibians comes from studies inferring multiple sexually-discordant genotypic female (XX♂) common frogs (*Rana temporaria*) from a single population (*Alho, Matsuba & Merila, 2010*) and for an individual sexually-discordant genotypic male (XY♀) common frog (*Rodrigues et al., 2018*). Key to resolving the role of environmental variation causing sex reversal are studies assessing sex reversal frequencies across multiple populations and in different ecological contexts (e.g., developed or undeveloped landscapes). In fact, almost a decade ago, *Wedekind (2010)* called for field-based research aimed at describing frequencies of sex reversal in

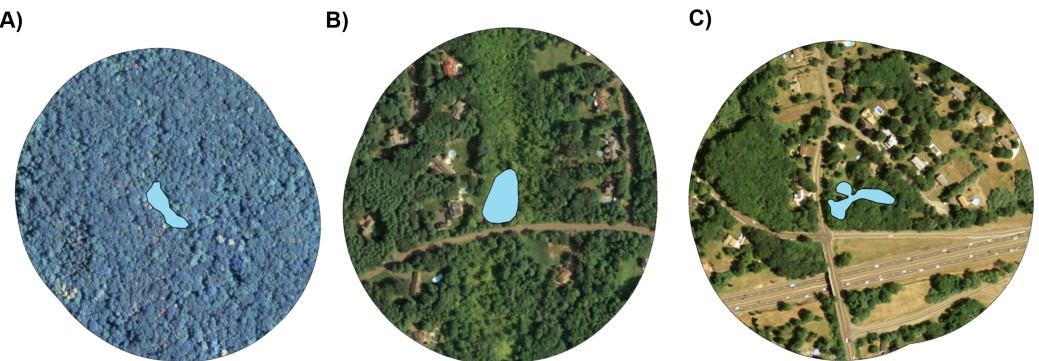

**Figure 1 Representative aerial images of three study ponds.** From left to right, ponds represent a varying degree of entirely forested with no residential land cover (A), an intermediate degree of suburban land cover (B), and a higher degree of suburban land cover (C). Ponds are depicted by blue polygons at the center of each image and land cover is shown within a 200 m radius of pond edges. Imagery is in the public domain and can be obtained from CT DEEP at http://www.cteco.uconn.edu/metadata/dep/document/ORTHO_2010_4Band_NAIP_FGDC_Plus.htm.

anthropogenic environments and environmental contexts with lower degrees of human activities to better untangle the extent to which sex reversal is related to natural vs anthropogenic causes. To our knowledge, no efforts to this effect have been made as of yet. Our study here aims to begin filling in this knowledge gap.

Green frogs provide a unique opportunity to study possible sex reversal in contrasting ecological conditions. We previously found that intersex green frogs (phenotypic male with testes exhibiting egg-like cells) commonly occur in suburban ponds (*Smits, Skelly & Bolden, 2014*) and that metamorphosing froglet phenotypic sex ratios vary along a forest-to-suburban land use gradient and are most skewed (biased toward phenotypic males) in forests (*Lambert et al., 2015*). Intersex and skewed sex ratios are regularly used as evidence for environmental sex reversal in amphibians, both as responses to temperature (*Witschi, 1914*, *1929*, *1930*; *Hsu, Yu & Liang, 1971*; *Dournon et al., 1984*; *Lambert et al., 2018*) and chemicals (*Hayes et al., 2002*; *Pettersson & Berg, 2007*; *Lambert, 2015*; *Lambert, Skelly & Ezaz, 2016*; *Tamschick et al., 2016a*, *2016b*).

We have also reported that, compared to forested ponds, suburban ponds harbor a diversity of contaminants known to influence sexual differentiation (*Lambert et al., 2015*). Three uncontaminated forested ponds and three suburbanized ponds from this prior work (*Lambert et al., 2015*) were studied here. Additionally, green frog tadpoles have a long larval period (ca. 1 year) and laboratory experiments show that green frog tadpoles can develop gonads of their predisposed genotypic sex early in larval development and then functionally sex reverse under certain environmental conditions, degenerating original gonads and developing the gonads of the opposite sex (*Mintz, Foote & Witschi, 1945*). By surveying green frogs along a forest-to-suburban land use gradient (Fig. 1; Fig. S1) and using a novel set of sex-linked molecular markers (*Lambert, Skelly & Ezaz, 2016*), we investigate the prevalence of environmental sex reversal across wild green frog populations and estimate the degree to which it represents a natural process or a response to human activities.

## METHODS

### Frog sampling

We focused on adult frogs to complement prior studies on intersex (*Murphy et al., 2006*; *Skelly, Bolden & Dion, 2010*; *Smits, Skelly & Bolden, 2014*), though are careful to distinguish GSD + EE here from sequential hermaphroditism which occurs in some fishes which sex reverse as adults (*Bachtrog et al., 2014*). Sexually-discordant adult frogs would have established their phenotypic sex opposite their genotypic sex prior to metamorphosis and not as adults. From April 25 to July 21, 2016 we collected adult phenotypic female and male green frogs from 16 ponds along a forest-suburban land use gradient (Fig. S2; Table S1). Nine ponds were studied in prior work (*Smits, Skelly & Bolden, 2014*; *Lambert et al., 2015*) and we recently acquired access to seven others. We collected frogs in buckets containing source pond water chilled on ice. Upon returning to the laboratory, we euthanized frogs with an overdose of buffered MS-222 and confirmed phenotypic sex via dissection and observation of the gonads. After euthanasia, we collected two samples of skeletal muscle from each frog in 95% ethanol and fixed each specimen in 10% buffered formalin. Pond and frog data are summarized in Table S1. In total, we collected 464 adult green frogs ($\mu$ = 29, range = 6–92) from the 16 study ponds. Of this sample, we collected a total of 129 phenotypic females ($\mu$ = 8.1, range = 2–17) and 335 phenotypic males ($\mu$ = 20.9, range = 4–75). Most phenotypic females were gravid, but a small proportion were spent, newly ovulating, or immature (Table S1). This study was approved by Yale IACUC protocols 2013-10361 and 2015-10681 and sampling was approved by CT DEEP Permit 0116019b.

### Land use classification

We identified ponds surrounded either entirely by undeveloped forests or by suburban neighborhoods (Fig. 1) using modeling analyses of high-resolution remotely-sensed imagery to classify the degree of suburban land use surrounding each of our study ponds (Fig. S3). To do this, we used a 2010 statewide orthophoto of Connecticut provided by the CT Department of Energy and Environmental Protection. This orthophoto was one-m resolution, was composed of four bands (infrared and natural color), and was taken during leaf-on (Fig. S3). We performed a supervised classification using the spatial analyst tools in ArcMap 10.1 (ESRI). Our targeted land cover types included trees, lawns, buildings, and roads or other paved surfaces. We supervised the classification with representative pixels (Fig. S2) surrounding each of the 16 ponds and estimated the percent cover of each land cover type. Ultimately, we merged lawn, building, and paved surface cover types into a single "Percent Suburban" predictor variable.

### Testicular histology

For histological analysis, we included all males from eleven ponds and randomly selected 22 males from the five remaining ponds with higher sample sizes ($\geq$26 males sampled). Following *Skelly, Bolden & Dion (2010)*, we removed the left testis from each male. At the Histology Core at the Yale School of Medicine, we retained every 20th section starting 10 sections into the gonad (*Skelly, Bolden & Dion, 2010*; *Smits, Skelly & Bolden, 2014*) and stained sections with hematoxylin and eosin. Each section was assessed for intersex

(the presence of oocytes) under a Leica BF200 compound microscope. We only included samples in our analyses which had at least four high quality sections. Sections were considered high quality if the staining was consistent and the gonad was predominantly intact.

## Genotyping

Tissue samples were preserved in 95% ethanol and transported to Diversity Arrays Technology Pty. Ltd. (Bruce, ACT, Australia) for genetic sequencing. We genotyped each frog at each of five recently identified sex-linked single nucleotide polymorphism (SNP) loci (RaclCT001, -003, -005, -007, and -009) (*Lambert, Skelly & Ezaz, 2016*) using DArTMP methods. For each locus, females are homozygous for the most common allele (i.e., X-chromosome allele) and males are heterozygous (i.e., have both a X- and Y-chromosome allele). Depending on the locus, we used either a multiplex or monoplex polymerase chain reaction (PCR) with locus-specific primers (Table S4) amplifying several base pairs around the sex-linked SNP locus. In the first round of PCR, the locus specific products were amplified in 30 cycles. In the second round of PCR, the sample-specific barcoded primers were used to identify individual frogs. The products of the second round of PCR were then sequenced on an Illumina Hiseq 2500.

## Genotypic sex assignments

Many anamniote taxa have homomorphic sex chromosomes which have traditionally made it challenging identify the sex chromosomes and the master SD gene for any given taxon (*Alho, Matsuba & Merila, 2010*; *Rodrigues et al., 2014*, *2018*). For amphibians, an entirely non-recombining master SD gene has been discovered in only one amphibian species (*Xenopus laevis*) to date (*Yoshimoto et al., 2008*, *2010*; *Kloc & Kubiak, 2014*). Through a variety of approaches (candidate genes, microsatellites, GBS), several studies have identified sex-linked genetic loci in a small number of amphibian taxa (*Berset-Brandli et al., 2006*; *Stock et al., 2011*; *Lambert, Skelly & Ezaz, 2016*; *Brelsford, Dufresnes & Perrin, 2016*; *Brelsford et al., 2017*). Individual sex-linked loci are likely to experience recombination and will therefore inaccurately identify genotypic sex at varying frequencies when analyzed individually. However, probabilistic Bayesian statistical methods are valuable techniques which allow us to statistically assign genotypic sex from multiple sex-linked genetic markers even in the absence of direct recombination rates at each locus from parent and offspring linkage analyses (*Alho, Matsuba & Merila, 2010*). By extension, such models provide inference into sex reversal by identifying sexual genotype-phenotype discordance, accounting for variation among loci in genotypic sexing reliability due to recombination. We take this approach in this study.

We used Bayes' theorem (below) to estimate the probability that each frog was a given genotypic sex and use Bayesian generalized linear models to estimate the probability that sex reversal was related to land use.

$$P(\text{Female} \mid \text{Allele combinations across loci})$$
$$= \frac{P(\text{Allele combination across loci} \mid \text{Female}) * P(\text{Female})}{P(\text{Allele combination across loci} \mid \text{Female}) * P(\text{Female}) + P(\text{Allele combination across loci} \mid \text{Male}) * P(\text{Male})}$$

We used prior combinations of genotypes across loci (Table S5) in the original set of 23 adult females and 54 adult males used to identify these sex-linked markers (*Lambert, Skelly & Ezaz, 2016*). We assumed the probability of being either sex was 0.5 but found that the value of P(Female) and P(Male) do not heavily influence the outcome of this analysis.

For the frogs here, we observed 31 different combinations of genotypes across loci for the sequenced markers. A total of 18 of these combinations were present in the original set of frogs used to identify the markers. These combinations were present in most sampled frogs in the study reported here. However, thirteen combinations, representing 88 individual frogs, were not previously observed. For these 13 combinations, we were not able to use prior combinations of genotypes. Instead, we used the prior frequencies of individuals with a given number of male and female genotypes for across all markers (e.g., an individual displaying a female genotype at three loci and a male genotype at two loci) regardless of the order of each locus. In most cases, the phenotypic sex and probable genotypic sex of these individuals were concordant.

Because of the possible effects of recombination influencing interpretation of sex reversal, we performed a second, but otherwise identical, analysis using just RaclCT001 and RaclCT003 (Table S6) because through our previous study we found that these two loci confidently assigned genotypic sex on their own (*Lambert, Skelly & Ezaz, 2016*). We constrained our analysis to individuals which had sequence data for both loci and individuals which had genotypes observed in our previous dataset. These criteria ultimately only excluded 13 phenotypic females and eight phenotypic males.

## Statistical analyses of sex

We analyzed putative associations between frequencies of sex reversal and intersex with land use within a Bayesian probabilistic framework following *McElreath (2015)* and using the maximum *a posteriori* (*map*) function in the R (v 3.4.0) package "rethinking." To do so, we modeled the probability of female-to-male sex reversal, male-to-female sex reversal, or intersex as binomial responses and used flat prior distributions for intercepts and land use slopes. For each the probability of female-to-male sex reversal, male-to-female sex reversal, and intersex, we developed two models. One model only included only an intercept term and the other include an intercept term and a term for suburban land use. We similarly modeled whether intersex frequencies were associated with female-to-male, male-to-female, or total sex reversal frequencies.

We evaluated performance among models using the widely applicable information criterion (WAIC) which takes the averages of log-likelihoods over the posterior distribution of a given model. WAIC values are particularly helpful inference metrics for evaluating Bayesian probabilistic generalized linear models because they do not rely on a Gaussian posterior distribution, therefore making them useful for binomial models like those used here. Furthermore, we assessed the relative predictive power of different models by comparing changes in WAIC (dWAIC in Table S3) and weighted model probabilities using Akaike weighted ranking (*McElreath, 2015*).

We note that, because sex reversal likely occurs during larval development and because the adult population represents multiple aggregated cohorts, it would be challenging to interpret any associations or lack of associations between sex reversal frequencies and environmental variables (e.g., temperature or pH) which may vary between years. Because of this, we limited our analyses to suburban land use which has remained a relatively constant measurement of the relative anthropogenic impact among populations over the past several years.

### Water chemistry

Our prior work demonstrated that the presence and diversity of endocrine disrupting chemicals capable of impacting vertebrate sexual differentiation are positively correlated with a suburban land use gradient (*Lambert et al., 2015*). Similarly, basic water quality parameters are also well known to be correlated with urban land use gradients (*Dow & Zampella, 2000*; *Conway, 2007*; *Zampella et al., 2007*; *Brans et al., 2017*), including in our own work (*Lambert et al., 2015*; *Holgerson et al., 2018*). As an additional layer to this study, we measured multiple water chemistry parameters to evaluate its association with the land use gradient as a relative measure of human influence among ponds. At most visits to each pond, we measured water conductivity (specific conductance), pH, and dissolved oxygen. For the first two measurements, we used an Oakton PCSTestr 35 Multiparameter probe. For dissolved oxygen, we used an YSI ProODO Handheld Optical meter. Conductivity in particular is a useful proxy for myriad contaminant sources entering urban ponds (*Dow & Zampella, 2000*; *Conway, 2007*; *Zampella et al., 2007*).

To characterize how land use might be associated with different water variables (conductivity, pH, dissolved oxygen, and temperature) we used linear models which included the percent of suburban land cover surrounding each pond as well as sampling date as predictor variables. If sampling date was not significant (at $p < 0.05$) in the model, we instead used linear mixed effects models with the "lmer" function in the *lme4* package in R, treating sampling date as a random effect to account for repeated measures, and treated suburban land use as the only fixed effect. For mixed effects models, we calculated an $R^2$ with the "r.squaredGLMM" function in the R package *MuMIn*.

## RESULTS

### Sex phenotype-genotype discordance

We inferred sex reversal from discordances between frog phenotypic sex and genotypic sex across five sex-linked markers, finding evidence that sex reversal occurs across most populations and in both directions. From 464 frogs (Table S1) collected across 16 populations, we observed sexual phenotype-genotype discordance in 4.5% ($n = 21$) of frogs and in 75% of populations (Fig. 2). Nine populations exhibited a single direction of sexual discordance while three populations exhibited bidirectional sexual discordance (Fig. 2). Across populations, 8.5% ($n = 11$, 89% credible interval 5.4–13.4%) of genotypic females had a male phenotype, whereas 3% ($n = 10$, 89% credible interval 1.8–4.9%) of genotypic males had a female phenotype.

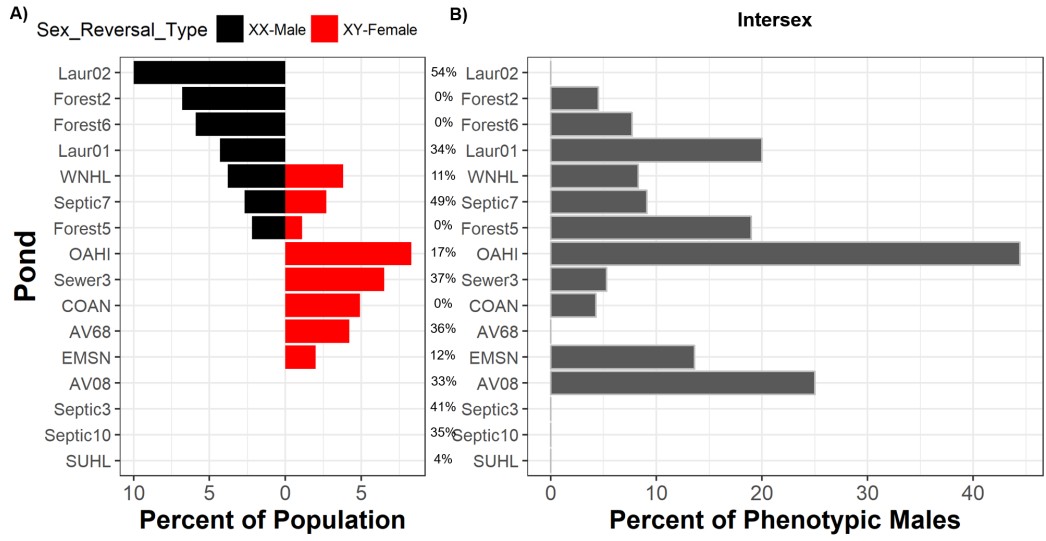

**Figure 2 Sex reversal and intersex are widespread but are not associated with suburban land use.**
Percent of each frog population comprised of sexually-discordant frogs (A) and percent of phenotypic males with intersex testes (B). Ponds are arrayed in descending order from greatest frequency of sexually-discordant genotypic females (XX-males; left bars) followed by the greatest frequency of sexually-discordant genotypic males (XY-females; right bars). Intersex and sexual discordance are not correlated. Percentages between panels indicate the percent of the landscape surrounding each pond that is comprised of suburban land use. Pond names match those in Table S1 and COAN, Forest2, Forest5, and Forest6 are forested. Table S1 also includes sample sizes.

Contrary to expectations that human land use drives sex reversal, we observed no such relationships between sex reversal frequencies, in either direction, and land use (Fig. S1). We detected sex phenotype-genotype discordant frogs in all uncontaminated forested ponds. Bayesian analyses indicated that suburban land use was not associated with frequencies of sexually-discordant genotypic females (XX♂; posterior probability = 0.500, 89% credible interval 0.495–0.510) nor sexually-discordant genotypic males (XY♀; posterior probability = 0.503, 89% credible interval 0.495–0.507) (Table S5; Fig. S1). In all cases, models only containing intercepts carried the majority of Akaike model weighting compared to models including suburban land use (Table S2). Similarly, WAIC values across models for either sex reversal type were relatively similar and standard errors of each models' WAICs were relatively high (Table S2). Importantly, changes in WAIC values (dWAIC) between the suburban model and the better model (lowest WAIC) with only an intercept were low. Additionally, the standard errors of estimated dWAIC were relatively high compared to actual dWAIC values. All metrics suggest no influence of land use on sex reversal. While it is possible that forested ponds studied here were exposed to some unmeasured contaminants, our previous work demonstrated that suburban land cover is a reliable proxy for the presence and extent of contamination in these ponds (*Lambert et al., 2015*).

To address the possibility that recombination could obscure the genotype-phenotype relationship leading to false positives for sex reversal, we performed a secondary analysis using a restricted dataset including only the most tightly sex-linked markers (those with alleles showing the strongest association with sex; see Methods).

This additional analysis provided qualitatively identical results (Table S3). Every individual identified as sex-reversed in this restricted analysis was previously assigned as sex-reversed in the original analysis. Of the 21 individuals excluded in this secondary analysis (see Methods for exclusion criteria), three had previously been assigned as sex-reversed and the remaining 18 were sex-concordant. No individuals present in both analyses were identified as sex-reversed in the first analysis but not in the second analysis. Across populations, 7% ($n = 8$, 89% credible interval 4.0–12.0%) of genotypic females displayed a male phenotype, whereas 3% ($n = 10$, 89% credible interval 1.8–5.0%) of genotypic males were phenotypically female. As with the prior analysis, we detected sexually-discordant frogs in all uncontaminated forested ponds. Model comparisons indicated a slight advantage of the model including suburban land cover (negative effect; posterior probability = 0.487, 89% credible interval 0.475–0.50) over an intercept-only model, suggesting a slight correlation between suburban land use and female-to-male sex reversal. However, the credible interval overlapped with 0.50, all model comparison metrics (Table S3) were only modestly different between the two models, and the proportional change in log odds was only 0.95 indicating the relationship between female-to-male sex reversal and suburban land use was minimal. Bayesian analyses for sexually-discordant genotypic males (XY♀) was not influenced by suburban land use (posterior probability = 0.503, 89% credible interval 0.495–0.507) (Table S3).

## Intersex

Of 246 phenotypic males histologically examined, 23 had intersex testes (9.3%, 89% credible interval 6.8–12.9%). Of the intersex frogs, sex-linked markers indicated that 91% ($n = 21$) were genotypically male while 9% ($n = 2$) were genotypically female, although a higher proportion of sexually-discordant phenotypic males (XX♂) were intersex compared to sexually-concordant phenotypic males (XY♂) (Fig. 3). While sexually-discordant phenotypic male (XX♂) frogs were limited in number, the posterior probability for genotypic sex influencing intersex frequencies was 0.28 (89% credible interval 0.10–0.60) and the proportional change in log odds of being intersex as a genotypic female (XX♂) compared to being a genotypic male (XY♂) was 0.40. This indicates that the odds of a sexually-concordant male (XY♂) being intersex is 60% lower than the odds for a sexually-discordant genotypic female (XX♂) such that 20% of sexually-discordant genotypic females (XX♂) were intersex whereas only 9% of sexually-concordant males (XY♂) were intersex. We note, though, that although XX♂ have a higher probability of being intersex, most intersex frogs were XY♂. Because most sexually-concordant males (XY♂) develop typical testicular morphology whereas many sex-reversed genotypic females (XX♂) likely retain oocytes in their testes, this result reflects differences in the frequency of XX♂ and XY♂ in wild populations.

We identified intersex phenotypic males in 10 of 16 populations (Fig. 2; Table S1). Half of ponds surrounded by suburban land cover had intersex males; for suburban ponds with intersex males, intersex frequencies ranged from 5.3% to 33.3% (Table S1). All four forested ponds harbored intersex males with intersex frequencies ranging from 4.3% to

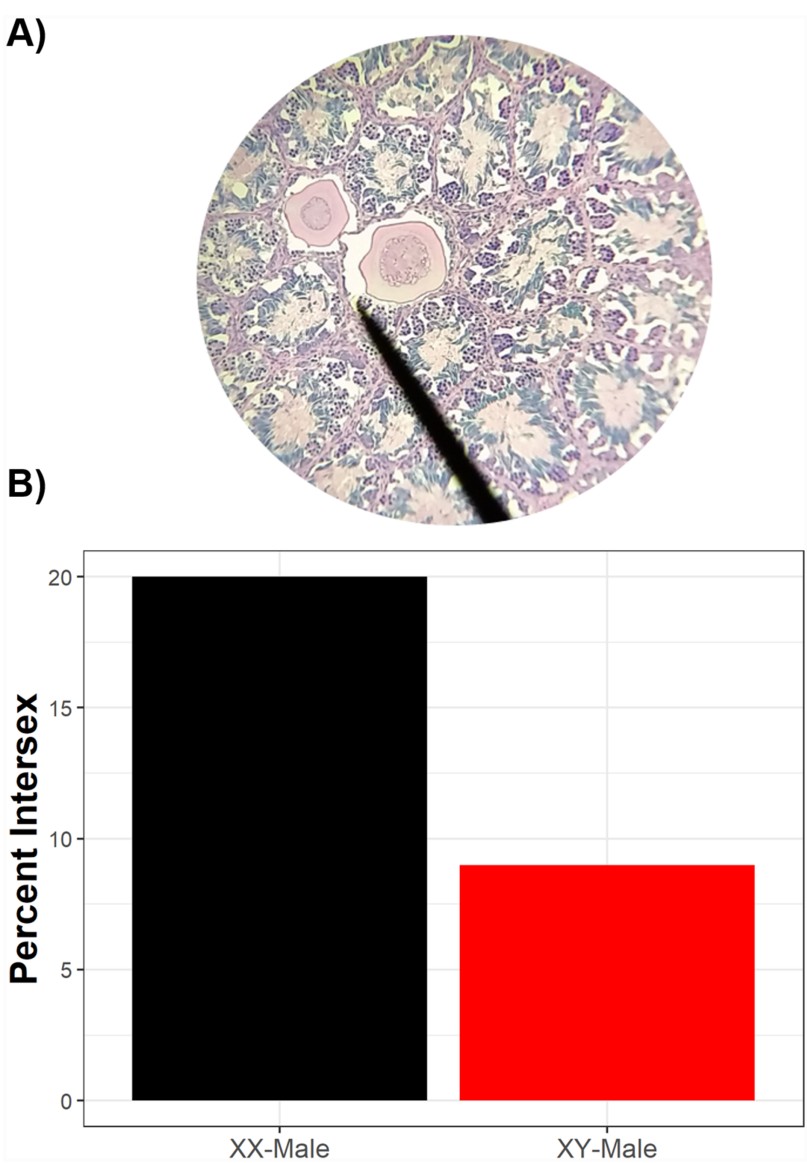

**Figure 3 Intersex phenotypic males can be either genotypic males or genotypic females.** Two testicular oocytes from an intersex green frog (A). While, overall, there are more genotypic males than genotypic females which are intersex (B), sexually-discordant genotypic females have a higher probability of exhibiting intersex than sexually-concordant males. Of phenotypic males histologically assessed, 10 were genotypic females and 236 were genotypic males.

19% of males in a given pond (Table S1). Bayesian models suggested intersex frequencies were not associated with human land use (posterior probability = 0.498, 89% credible interval 0.493–0.503; Table S2; Fig. S1). WAIC and dWAIC standards errors were again high and Akaike weighting again did not favor the model including suburban land use (Table S2; Fig. S1). This result is mirrored by data (H.E. Bement & D.K. Skelly, 2011, unpublished data) showing equal intersex frequencies in green frog populations from each of three forested (range 6.7–25% intersex) and three suburban ponds (range 4.3–17% intersex). Models also indicated that intersex frequencies were not associated with rates

of female-to-male (posterior probability 0.49, 89% credible interval 0.46–0.52), male-to-female (posterior probability 0.53, 89% credible interval 0.49–0.56), or total sex reversal frequencies (posterior probability 0.51, 89% credible interval 0.47–0.54).

## Water chemistry

Dissolved oxygen, pH, conductivity, and temperature all increased with increasing suburban land cover surrounding ponds. The model (full model $R^2 = 0.29$, $p < 0.001$) for dissolved oxygen included both Percent Suburban ($p = 0.002$, estimate = $0.06 \pm 0.02$) and Sampling Date ($p < 0.001$, estimate = $-0.07 \pm 0.02$). Dissolved oxygen generally declined throughout the season by 0.7 mg/L every 10 days and, controlling for Sampling Date, dissolved oxygen increased by 0.6 mg/L for every 10% increase in landscape composition that was suburban land cover.

For pH, the model (full model $R^2 = 0.22$, $p < 0.001$) included both Percent Suburban ($p < 0.001$, estimate = $0.02 \pm 0.003$) and Sampling Date ($p = 0.04$, estimate = $0.006 \pm 0.003$). Generally, pH increased throughout the season by 0.06 every 10 days and, controlling for Sampling Date, pH increased by 0.2 for every 10% increase in suburban land cover composition.

In the linear regression (full model $R^2 = 0.44$, $p < 0.001$) for conductivity, Percent Suburban was significant ($p < 0.001$) but Sampling Date was not ($p = 0.10$). A linear mixed effects model treating Sampling Date as a random effect found a significant positive correlation between conductivity and the fixed effect Percent Suburban ($R^2 = 0.43$, $p < 0.001$, estimate = $6.5 \pm 0.9$). Conductivity increases on average by 65 microsiemens for every 10% increase in landscape comprised of suburban land cover.

For temperature, the model (full model $R^2 = 0.52$, $p < 0.001$) included both Percent Suburban ($p < 0.001$, estimate = $0.06342 \pm 0.01528$) and Sampling Date ($p < 0.001$, estimate = $0.12419 \pm 0.0141$). Generally, temperature increased throughout the summer by 1.2 °C every 10 days throughout the summer and, controlling for Sampling Date, temperature increased by 0.6 °C for every 10% increase in landscape composition that was suburban land cover.

These data help reconfirm the use of our suburban land use gradient as a proxy for contamination. Chemical data from forested and suburban pond studied here and in prior work can be found in *Lambert et al. (2015)*.

## DISCUSSION

Our study suggests that sex reversal in green frogs may be common and frequent. By identifying sex reversal from discordances between phenotypic and genotypic sexes inferred across sex-linked markers, we found evidence for sex reversal in 12 of 16 populations. Interestingly, our analyses suggest sex reversal occurs independently of human land use and occurred in all undeveloped, forested frog populations. While we cannot discount possible contamination from unmeasured sources, our recent chemical analyses have demonstrated that our suburban land use gradient is a reliable proxy for the presence, diversity, and concentration of contaminants which can influence sexual development (*Lambert et al., 2015*). Importantly, our evidence to date indicates minimal

or no detectable contamination in the types of forested ponds we studied, including three of the four forested ponds studied here. Sexual development in green frogs is likely a complex process whereby some individuals develop sexually-concordant, some develop sexually discordant early in larval development, and some begin developing as their predisposed genotypic sex but sex reverse into the opposing phenotypic sex later in larval development but prior to metamorphosis. Our results supports earlier findings from European common frogs (*Alho, Matsuba & Merila, 2010*; *Rodrigues et al., 2014*) to suggest that the environment plays a greater role in determining sexual trajectories in amphibians than is typically discussed (*Miura, 2017*; *Sarre, Ezaz & Georges, 2011*; *Evans, Alexander Pyron & Wiens, 2012*; *Bachtrog et al., 2014*; *Capel, 2017*) and may not necessarily be a response to anthropogenic conditions. Our findings here elucidate new patterns of sex reversal and intersex in wild frogs and highlight areas of future research needed to more clearly identify underlying mechanisms as well as the ecological and evolutionary ramifications of sex reversal and intersex.

## Potential causes of sex reversal

The bidirectional sexual discordance observed here perhaps indicates that green frog sexual development may be influenced by multiple factors. Laboratory experiments show the direction of amphibian sexual differentiation away from the genotypically predisposed sex (e.g., genotypic female to phenotypic male or genotypic male to phenotypic female) can vary by species, demes, and environmental conditions (*Miura et al., 2016*; *Tamschick et al., 2016a*). The bidirectional sex reversal observed here in adult green frogs could possibly indicate that amphibian sexual differentiation is guided by multiple environmental (temperature and chemical) factors or yearly variation in the environment. If this were true, it may be challenging to identify specific mechanisms of sex reversal. Unfortunately, green frogs require 2–3 years post-metamorphosis to mature and survive for up to 7 years (*Shirose & Brooks, 1995*) and so we are unable to assess sex reversal frequencies as a function of a single year's water conditions in the adults sampled here due to multiple overlapping cohorts. Additionally, we cannot discount that frogs may have moved from the environment which led to their intersex or sex-reversed condition before they were collected. Additional studies assessing patterns in environmental conditions with sex reversal frequencies in wild cohorts of larval or metamorphosing amphibians for which the collection site and the larval environment are known to be the same will be critical for identifying putative drivers (e.g., temperature, pH, dissolved oxygen, organic chemicals, etc.) of sex reversal as well as contributions to cohort sex ratio variation.

## Intersex

While we identified both genotypically female and male intersex frogs, intersex and sexual genotype-phenotype discordance frequencies were not correlated, perhaps suggesting the mechanisms underlying intersex and sex reversal may be different. By extension, this suggests that we cannot conflate intersex and sex reversal as has been suggested previously (*Murphy et al., 2006*). The few experiments which have assessed both

intersex and genotypic sex have not reported relationships between sex reversal and intersex (*Hayes et al., 2010*; *Tamschick et al., 2016a*, *2016b*). Future experimental and field studies should assess any relationship between intersex and sex reversal to better understand the conditions and mechanisms underlying these processes. We note that intersex is likely a terminal phenotype in most amphibians and, at least in green frogs, is not evidence of sequential hermaphroditism which currently is not known to occur in amphibians. Phenotypic males with intersex testes may represent two developmental scenarios. In one pathway, larval genotypic females either partially or completely develop their ovaries but are later environmentally-induced to sex reverse, degenerating their ovaries and developing testes prior to metamorphosis. This pathway has been experimentally demonstrated in green frog tadpoles previously (*Mintz, Foote & Witschi, 1945*). Alternatively, genotypic males may experience abnormal gonadal differentiation but do not flip their sexual trajectories toward being phenotypic females (*Tamschick et al., 2016a*, *2016b*). Both field-based and experimental research could provide exciting insight into the relative contribution of these two developmental pathways to intersex frequencies in the wild.

In recent years, discussions about amphibian sexual development have often been guided by striking results from laboratory-based ecotoxicological studies which have led to the impression that sex reversal and intersex are predominantly the result of human factors (*Hayes et al., 2002*, *2010*; *Pettersson & Berg, 2007*; *Tamschick et al., 2016a*, *2016b*). However, previous work in other frog species throughout the past century (*Witschi, 1914*, *1929*, *1930*; *Hsu, Yu & Liang, 1971*) including a field survey in Africa (*Du Preez et al., 2009*), also indicates that intersex can be widespread among wild populations and may not be necessarily related to contaminants but rather natural environmental factors like temperature. It currently remains unclear what environmental conditions cause intersex to develop in green frogs and other taxa. Our findings that sexual genotype-phenotype discordance and intersex frequencies were not associated with land use is therefore intriguing and perhaps surprising. These results should prompt a reconsideration of the perception that these sexual characteristics are predominantly of anthropogenic origin.

## Evolutionary and ecological implications

Despite having experienced at least 32 transitions between GSD (e.g., XY vs ZW sex chromosome) systems (*Evans, Alexander Pyron & Wiens, 2012*), amphibians are not generally believed to naturally undergo sex reversal (but see *Perrin, 2009*; *Alho, Matsuba & Merila, 2010*; *Rodrigues et al., 2018*). Yet, most mechanistic models of sex determination transitions invoke a role for environmentally-sensitive sex determination (*Bull, 1981*; *Grossen, Neuenschwander & Perrin, 2011*; *Quinn et al., 2011*; *Schwanz et al., 2013*; *Muralidhar & Veller, 2018*). Our findings in conjunction with research on European common frogs should encourage future investigation in to whether sex reversal influences transitions among amphibian SD modes.

Numerical modeling suggests that the frequency of sexually-discordant genotypic females (XX♂) reported here should have minimal effects on population size

or viability unless sexually-discordant genotypic females (XX♂) have reduced fitness (*Cotton & Wedekind, 2009*; *Wedekind, 2010*). The low frequency of sexually-discordant genotypic males (XY♀) in green frogs, however, may permit the type of sex-chromosome rejuvenation suggested by the "fountain of youth" hypothesis (*Perrin, 2009*; *Grossen, Neuenschwander & Perrin, 2012*).

## Sex chromosome recombination

The main complication with identifying sex reversal in taxa like amphibians, fishes, and non-avian reptiles is having to identify and rely on sex-linked genetic loci which may or may not be the SD locus. While "the SD locus" should experience no recombination, other loci on the sex chromosomes will experience varying degrees of recombination depending on their proximity to the SD locus as well distance between the loci and the centromere. Therefore, while a number of loci may all be sex-linked based on their corresponding location to SD locus, certain individual sex-linked loci will show different degrees of recombination and therefore sex-linkage. Because of this heterogeneity in linkage, alleles associated with the heterogametic sex chromosome (e.g., Y or W) will possibly be observed in the opposite genotypic sex and the heterogametic sex will occasionally exhibit the homogametic sex's genotype at certain sex-linked loci (*Hussain et al., 1994*). Recombination rates for sex-linked loci can be directly estimated with family inheritance data from both parents and their offspring (*Matsuba, Miura & Merila, 2008*). While having directly estimated recombination rates would be useful to our study, we are not prevented from identifying informative sex-linked loci in their absence (*Lambert, Skelly & Ezaz, 2016*; *Brelsford et al., 2017*).

Bayesian probabilistic modeling approaches like the one used here and first used by *Alho, Matsuba & Merila (2010)* to infer sex reversal in European common frogs are useful tools for inferring the genotypic sexes of individuals across multiple loci by accounting for variation in sex-linkage and, by extension, recombination. As we discussed previously (*Lambert, Skelly & Ezaz, 2016*), the sex-linked loci identified for green frogs show varying degrees of sex-linkage, likely due to recombination. RaclCT001 is a particularly interesting locus because its alleles were perfectly sex-linked (all females were homozygous for the common X-allele and males were heterozygous) across 23 adult females and 54 adult males in the original dataset (*Lambert, Skelly & Ezaz, 2016*). And in this current study of over 400 frogs, RaclCT001 was again perfectly sex-linked outside of inferred sex-reversed individuals suggesting that it is either at or very near the SD locus in these populations and experiences minimal if any recombination. By analyzing across multiple loci, and in particular locus RaclCT001, we can accurately assign the genotypic sex to green frogs in our study population even if multiple loci for an individual are apparently discordant. Although we cannot fully discount the possibility that sex chromosome recombination influenced our results, the methods we employed account for variation in sex-linkage and allow for useful inference into individuals' genotypic sex, and therefore sex reversal, in the absence of a well-established SD locus. Recombination likely accounted for variation in sex-linkage at several of the loci (RaclCT005, -007,

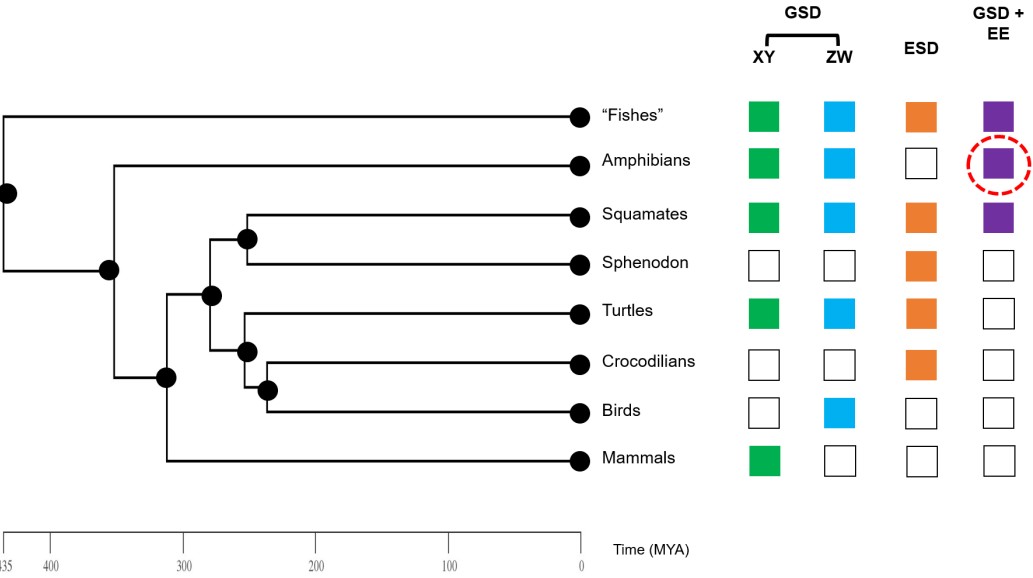

**Figure 4 Phylogeny of vertebrate sex determination, now including sex reversal in amphibians.** Genotypic sex determination (GSD), environmental sex determination (ESD), and sex reversal (GSD + environmental effects (EE)) are variable across the vertebrate phylogeny. Amphibians were previously generally thought to only exhibit strict GSD. We provide early data here (red dashed circle) that environmental sex reversal may be a natural process in amphibians.

and -009) used here but was unlikely to be so extreme to as result in our analyses mis-assigning genotypic sexes. Our analyses using either the complete sex-linked markers set or only the most tightly sex-linked loci produced similar results, validating our interpretations of sex reversal. However, future work in this system using family data will be useful for unequivocally disentangling the effects of recombination in these markers, if there is any.

## CONCLUSIONS

By inferring sex reversal in green frogs living in different anthropogenic and undeveloped ecological contexts, we provide early support that sex reversal might be a relatively natural and frequent process in this species. Our study highlights the need for more research on amphibians in various ecological contexts and for biologists studying sex reversal and sexual development to report some metric of the ecological conditions of the populations they study. In recent years there has been a substantial focus on contaminant-caused intersex and sex reversal. Yet there are few studies assessing intersex or sex reversal in replicate undeveloped landscapes, particularly as a contrast to developed landscapes with known contamination. As called for almost a decade ago by *Wedekind (2010)*, such data are critical for disentangling the extent to which sex reversal is of anthropogenic or natural origins. Additionally, our work should spark interest in understanding how well "control" conditions in laboratories represent baseline conditions for sexual development. Even in undeveloped landscapes with minimal contamination, amphibian larvae develop in heterogeneous environments comprised of myriad

natural chemicals as well as varying temperature regimes. Therefore, sex reversal could very well be an outcome of multi-stressor response (including temperature). While our study focused on adult frogs, and therefore cannot draw conclusions about putative drivers of intersex or sex reversal, future research focused on sexual development in larval and metamorphosing amphibians in the wild could help identify any natural or anthropogenic mechanisms of sex reversal and the development of intersex.

Our study shines new light on sex reversal in an amphibian species and begins to expand our understanding of sex reversal in amphibians relative to other vertebrates (Fig. 4). Even so, our study was limited to a single species and in a relatively constrained region of its large range. Future investigations into patterns of environmental sex reversal in other species, particularly from diverse amphibian taxa including salamanders and caecilians, as well as various ecological contexts are paramount for understanding the evolutionary and ecological drivers and consequences of dynamic amphibian sexual development. Concentrated efforts in recent years coupled with modern genomics tools have illuminated an astonishing diversity of SD modes in fishes and squamates (*Pennell, Mank & Peichel, 2018*). We are now ready for a focused investigation of the sexual diversity of living amphibians.

## ACKNOWLEDGEMENTS
Lauren Savidge helped identify new frog populations and Os Schmitz, Gunter Wagner, Adam Roddy, Geoff Giller, Rob Buchkowski, and Mark Bradford improved this manuscript. Greg Watkins-Colwell facilitated this research. We thank homeowners who provided access to their yards.

### Funding
Funding came from the Garden Club of America and Yale Institute for Biospheric Studies to Max R. Lambert, the Yale STARS program to Tien Tran, an Australian Research Council Future Fellowship (FT110100733) to Tariz Ezaz, and the Peabody Museum to David K. Skelly. The funders had no role in study design, data collection and analysis, decision to publish, or preparation of the manuscript.

### Grant Disclosures
The following grant information was disclosed by the authors:
Garden Club of America and Yale Institute for Biospheric Studies.
Yale STARS program.
Australian Research Council Future Fellowship: FT110100733.
Peabody Museum.

### Competing Interests
Andrzej Kilian is the founder and director of Diversity Arrays Technology. The authors declare no competing interests.

## Author Contributions

- Max R. Lambert conceived and designed the experiments, performed the experiments, analyzed the data, contributed reagents/materials/analysis tools, prepared figures and/or tables, authored or reviewed drafts of the paper, approved the final draft.
- Tien Tran performed the experiments, analyzed the data, authored or reviewed drafts of the paper, approved the final draft.
- Andrzej Kilian performed the experiments, contributed reagents/materials/analysis tools, authored or reviewed drafts of the paper, approved the final draft.
- Tariq Ezaz conceived and designed the experiments, contributed reagents/materials/analysis tools, authored or reviewed drafts of the paper, approved the final draft.
- David K. Skelly conceived and designed the experiments, contributed reagents/materials/analysis tools, authored or reviewed drafts of the paper, approved the final draft.

## Animal Ethics

The following information was supplied relating to ethical approvals (i.e., approving body and any reference numbers):

Yale IACUC approved this work (protocols 2013-10361 and 2015-10681).

## Field Study Permissions

The following information was supplied relating to field study approvals (i.e., approving body and any reference numbers):

The Connecticut Department of Energy and Environmental Protection (CT DEEP) approved this work (Permit 0116019b).

## Data Availability

All specimens, tissues, and histological sections are deposited in the Yale Peabody Museum of Natural History (YPM HERA 015904-015992, 018641-018672, 018913-019255, 019666). Extended methodology, figures, tables, and data are available as Supplementary Material (Figs. S1–S3; Tables S1–S6, Dataset).

## Supplemental Information

Supplemental information for this article can be found online at http://dx.doi.org/10.7717/peerj.6449#supplemental-information.

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
