# Peer review of "Molecular evidence for sex reversal in wild populations of green frogs (Rana clamitans)"

_PeerJ, doi:10.7717/peerj.6449_

## Round 0.1 · original submission · Major Revisions

The submitted manuscript details both molecular and phenotypic evidence of sex reversal and intersex, not explicitly linked with land-use. The authors suggests that this could mean that genetic sex determination in amphibians is influenced by environmental factors. I agree that there is some evidence to support this hypothesis. The data in some cases are compelling and the approaches used are appropriate. However, as mentioned by one of the reviewers, some of the conclusions seem to be a bit speculative.

I encourage the authors to thoroughly address reviewer comments and concerns, with a particular focus on shortening up the discussion (as another reviewer said it is a bit on the long side) and being cautious in making conclusions that aren't fully supported, as this is one data set. For example, why is the direction of sex reversal and intersex not similar across ponds if land use doesn't matter? It would be nice to make the lack of trends a little clearer in figures by splitting the site names more clearly between forested and suburban perhaps with an (F) and (S). Also, I agree with the comment that there should be some type of statistical comparison indicated on Figure 3. Another overall editorial comment is that in some cases very general statements are made in which the only citation is Lambert et al. 2015 or 2016, for example on defending the use of water qualities such as pH and temperature to classify ponds as impacted or not. Other supporting literature should be included too.

Reviewer 1 ·

Basic reporting

No commets. Comments are in the general report.

Experimental design

No commets. Comments are in the general report.

Validity of the findings

No commets. Comments are in the general report.

Additional comments

This is an interesting work on the study of the relations between phenotypic and genotypic sexes of an amphibian species in the wild.
However, I do not clearly see why they think that sex reversal might be a relatively natural and frequent process in this species. If so, why the direction is not the same across the different ponds?


Major points:

Figure one: It is clear that in some ponds: Laur02, Forest2, Forest6 and Laur01 the authors only found XX-Males and in other ponds, only XY females. I did not find in the text a clear or at least some explanation for this behavior. This is also true for the ponds of both sex reverse directions were found.

Figure two: Same considerations for the presence of intersex gonads.

Sex determination markers: Is it not possible that the specific sex markers the authors used are not 100% reliable? Thanking into consideration that some genes have been proposed to be involved in the sex determining system in amphibian such as DM-W. I understand there is a diversity of sex determining mechanisms, but it would be nice if the authors include a discussion on this respect.


Minor points:

Introduction. Page 3. Lines 67-68. The citation Yoshimoto et al. 2008 is in the reference section but not in the text. It should be here. I would also consider that the authors should have to add here another citation from the same group: Yoshimoto et al. 2010. Development 137, 2519-2526. 2010.

Methods. Page 8. Line 179. Please change Yoshomoto by Yoshimoto.

References. Page 31. Line 701. Please change “200” by “2008”.

Reviewer 2 ·

Basic reporting

Meets requirements. See General Comments below.

Experimental design

Meets requirements. See General Comments below.

Validity of the findings

Meets requirements. See General Comments below.

Additional comments

This is important work which starts to investigate both the ecological and environmental context of sex reversal in amphibians. Overall I thought this was a well-constructed piece of work. The authors clearly state their hypotheses, and explain how their work fills an important gap in the knowledge. I really appreciated that is work is empirically testing some long-held assumptions in the field – that sex reversal is an anthropogenic effect rather than a naturally occurring phenomenon. I also appreciated that the authors spent time clearly disambiguating two often conflated phenotypes: intersex from true sex reversal.

I have no major corrections, just a question. In frogs, is intersex a terminal or transitional phenotype? Perhaps the authors can further clarify that intersex testes are not evidence of sequential hermaphroditism? (This was mentioned but the paper could clarify this a little more). If intersex is a terminal phenotype, can the authors comment on whether these intersex individuals are likely to be fertile and capable of successfully reproducing as one sex or another?

Minor typographical corrections:

Line 93: “anthropogenic environments and environments contexts with lower degrees of human activities”. - should be “environmental contexts”.
Line 215 : “These criterial ultimately” – should be “criteria”.

Reviewer 3 ·

Basic reporting

In this study, Lambert et al assess the prevalence of sex reversal and intersex in adult green frogs (Rana clamitans) across 16 wild populations sampled along a forest-to-suburban land use gradient. By comparing phenotypic sexes inferred by histological analysis and genotypic sexes inferred by five sex-linked SNP loci, they found evidence for sex reversal in 12 of 16 populations, including all the four populations inhabiting ponds in undeveloped forests. By examining the presence of oocytes in phenotypic male testis, they found that 10 of 16 populations had intersex phenotypic males, and XX-males have higher probability being intersex than XY-males. They also applied Bayesian analyses to examine the associate between land use and the occurrence of sex reversal or intersex, and they found no influence of suburban land use on sex reversal or intersex frequencies. In general, the manuscript is well organized, and the methods are sound. Given that there are few field-based studies assessing sex reversal in amphibians in contrasting ecological conditions (natural vs anthropogenic), the results from this study are important for this field, providing early evidences for understanding the extent to which sex reversal is related to natural or anthropogenic causes in amphibians.

Experimental design

no comment

Validity of the findings

no comment

Additional comments

Throughout the manuscript, the authors frequently use the terms “sexually-discordant phenotypic male”, “sexually-concordant phenotypic males”, “sexually-discordant genotypic female” et al to represent sexual several or normal frogs, which sometimes are quite fuzzy, particularly the authors sometimes use different terms to represent the same meaning, such as sexually-discordant phenotypic male vs sexually-discordant genotypic female. I would suggest the authors add symbols like XX-male, XX-female, XY-males or XY-female in parentheses after each term, e.g. “sexually-discordant phenotypic male (XX-male)”, so that readers can readily understand the meaning of these terms without imaging the genotype and phenotype in their brains.

In the abstract, the authors concluded that “our results also suggest that intersex phenotypic males and sex reversal occur independently of each other”. But according to Line 313-319 and Figure 3, XX-males show twice the probability of being intersex compared to XY-males. Doesn’t it mean that intersex is associated with female-to-male sex reversal?

Figure 2: Given that the sample sizes vary greatly among the 16 populations (ranging from 6 to 92), I suggest the authors present the total number of frogs for each pond in this figure, so that readers can judge the reliability of the frequency for each population. In addition, the symbols of the ponds are meaningless for reads, thus I also suggest the authors show the percent of suburban land use for each pond, so that readers can readily find out the ponds entirely surrounded by undeveloped forests or those heavily surrounded by anthropogenic lands.

Figure 3: the bar plot shows that XX-males have much higher probability being intersex than XY-males. However, the statistical significance of this difference was not assessed. Given that the sample size for XX-male (only 11 in total) is quite small, this observed difference may be insignificant and resulted from random effect.

L301-304: It is unclear if the authors want to say suburban land use was slightly associated with frequencies of sexually-discordant genotypic females here. Please improve the description and indicate your finding more explicitly to readers.

In many of the correlation analyses, it would be easier to follow if the authors could provide the plots.

I would suggest the authors summarize the results in the subtitles. And the discussion is very lengthy making it hard to follow the purpose for some of the discussion. For instance, the authors basically did not provide any new clue on the mechanisms for sexual reversal. The discussion is quite redundant and verbose. Also some of the contents in the discussion are unnecessary background that not related with this work. I would suggest the authors to shorten the discussion section and more focus on their current results.

I cannot find the figure legends for all the supplemental figures.

---

## Round 0.2 · accepted · Accept

Thank you for taking the time to thoroughly address reviewer and editorial comments. This is an interesting paper that will contribute to a better understanding of the variability in amphibian sex determination.

# Reviewer 1 ·

Basic reporting

No comment

Experimental design

No comment

Validity of the findings

No comment

Additional comments

This is a nice piece of work. Congratulations.